# The Moderating Role of Maternal Education and Employment on Child Health in Pakistan

**DOI:** 10.3390/children9101559

**Published:** 2022-10-14

**Authors:** Muhammad Farhan Asif, Shafaqat Ali, Majid Ali, Ghulam Abid, Zohra S. Lassi

**Affiliations:** 1National College of Business Administration and Economics, Lahore 54000, Pakistan; 2Department of Statistics, Kohsar University, Murree 43600, Pakistan; 3Department of Economics and Agri. Economics, PMAS-UAAR, Rawalpindi 43600, Pakistan; 4Kinnaird College for Women, Lahore 54000, Pakistan; 5Robinson Research Institute, University of Adelaide, Adelaide, SA 5005, Australia

**Keywords:** moderating role, maternal education, employment status, unmet need for family planning, child health, Pakistan

## Abstract

Background: Pakistan has challenges in fulfilling its universal responsibilities of providing better health facilities to everyone. The Sustainable Development Goals (SDGs) aim to reduce maternal and infant mortality rates. Despite declines in mother and child death, the total mortality ratio has marginally increased. However, neonatal death has not decreased significantly. Family planning is important for controlling population growth and improving child as well as maternal health. Pakistan’s government has unceasingly tried to enhance the provision of contraceptive facilities, but still, an unmet need for family planning (UMNFP) exists in our country. Women are said to have UMNFP if they want to limit or space childbearing, but they are not using contraception methods for any reason. The study aimed to explore the effect of the UMNFP and to investigate the moderating role of a mother’s education and employment status on a child’s health. Methods: We analyzed the data of 2,244 women in this study. To investigate the study objectives, we utilized the secondary dataset of the Pakistan Demographic and Health Survey (PDHS) 2017–18 (publicly available on the website of the National Institute of Population Studies) and applied binary logistic regression using SPSS 24. Results: Results suggest a positive effect of a woman’s age (25 to 39 years), maternal education (higher), father’s education (higher), family’s wealth status (richest), mass media exposure, and adequate birth spacing (at least for 33 months) on a child’s health. On the other hand, there is an indirect association between maternal employment, unmet need for family planning, and a child’s health. The moderating role of maternal education and employment on the relationship between household wealth status and a child’s health is positive. Conclusions: We conclude that the strong predictors of child health are UMNFP, maternal education, and employment. The link between the met need for family planning and the child’s health is positive. The moderating effect of maternal education and household wealth status on a child’s health is progressive. Similarly, the interaction effect of a mother’s employment and household wealth status on a child’s health is positive. Finally, we concluded that the link between the health of the child and household wealth status is much more diverse and positive when the mother is highly educated and currently employed.

## 1. Background

Rapid population growth is one of the important characteristics of low- and middle-income countries (LMICs), and Pakistan is no different. Pakistan ranks sixth among the most populous countries in the world, and two thirds of the population lives in the rural and remote parts of the country [1]; in the year 2050, Pakistan will rank fifth on the list of the most populous countries in the world [2]. Human capital, i.e., literacy and skills, are important determinants of economic development. However, the literacy rate is 49% (56% in males vs. 43% in females) in people of 15 to 45 years of age. 

The maternal mortality ratio (MMR) in Pakistan has dropped from 521 per 100,000 live births in 1990 to 178 per 100,000 live births in 2019. Postpartum hemorrhage, eclampsia, obstructed labor, and sepsis are the major causes of high MMR in LMICs, and 20% of the 8000 deaths per annum in women aged 15–49 in Pakistan [1,3,4]. At the same time, variations in MMR exist between rural and urban areas, i.e., high MMR in rural areas (319 per 100,000 live births) as opposed to urban regions (175 per 100,000 live births). Similarly, variations exist between geographical settings, i.e., the highest is in the province of Baluchistan (298 per 100,000 live births) as opposed to Punjab (157 per 100,000 live births) [3].

Children are viewed as a country’s future human resource, and this is universally acknowledged [5]. For the betterment of a nation, then, children’s health must be firmly assured. Despite considerable breakthroughs in medical technology and administration, governments continue to struggle to reduce the mortality and morbidity rates of children [1]. Regarding under-5 child mortality rates, Pakistan currently ranks 25th in 225 countries worldwide [6]. Although the child death rate has reduced from 141 per 1000 live births in 1990 to 67.2 per 1000 live births in 2019 [7], it is far slower than the Sustainable Development Goals of reducing it to 46 per 1000 live births by 2030 [6,8]. Almost half of the child deaths occur within 30 days of birth (202,000/year) and these are mostly because of diarrhea, pneumonia, and malaria [9]. These deaths are higher among those who are malnourished or less weight at the time of birth. According to recent estimates, the important reasons for mortality throughout the postnatal period are pneumonia (26%) and diarrhea (27%); and these are narrowly interlinked with poverty, under-nutrition, and poor sanitation and hygiene [3,8].

In Pakistan, the impact of social determinants affecting maternal and child health cannot be undervalued. The socioeconomic determinants and women’s education, region of residence, and wealth status. Like maternal mortality, there are geographical and rural–urban variations in terms of child mortality [3,6]. Child mortality is 2.5 times higher (119/1000 live births) in poorer families compared with wealthier families (48/1000 live births). Similarly, the death rates are higher in the province of Baluchistan (111/1000 live births) than in the province of Khyber Pakhtunkhwa (70/1000 live births) [3].

Socioeconomic status (SES), which is directly proportional to a family’s income, is one of the most influential factors of child health. According to the literature, children from less affluent households encounter a variety of health challenges, including accidents, lung diseases, etc. [9,10,11]. On the contrary, children from affluent homes are able to meet the expenses of good quality diets and sanitary services that protect them from developing health issues [12]. Children’s health is not just determined by income; many other variables of socioeconomic status have been studied in prior research [13]. Education plays a greater role than income in lowering child health difficulties [14]. Mothers with higher levels of education are more knowledgeable about and concerned about the health of their children [15]. In addition, the level of education held by both parents is linked to the likelihood that their children may suffer from diseases, infections, and inadequate nutrition [16]. The mortality in children whose mothers have received no formal education is two-fold higher (112/1000 live births) than in children whose mothers have received education up to secondary level (57/1000 live births) and three-fold higher than in children whose mothers have received education higher than secondary level (36 per 1000 live births) [17]. The relationship between women’s employment (MES) and their use of maternity and child health services has been the subject of a variety of contradictory studies. According to certain research demonstrating a positive correlation, women’s employment increases their control over earned income, hence empowering them to seek maternity and child health services [18]. In the same way, a child is more probable to have health problems or die if the health of the mother is weak during pregnancy [19]. Access to the media has a significant impact on enhancing the health of mothers and children. Children must be supplied with vaccinations and adequate health care facilities, as presented on television programs. Mothers can better care for their children as a result of this insight [20]. Birth interval and child’s health also contribute to child mortality; birth spacing between two and four years increased newborn subsistence by 2.4 times and child subsistence by 2.9 times [21]. Similarly, neonatal death among those children who are small at birth is double as compared with those infants who are average or large at birth [22].

Unmet need for family planning (UMNFP) also impacts a child’s health. UMNFP results in close birth intervals at a very young age [23], which sometimes causes abortion and poor maternal health, which are considered major contributors to poor child health, and high maternal and child death [24]. This paper has utilized the dataset from the Pakistan Demographic and Health Survey (PDHS) 2017–2018 to examine the effect of UMNFP on a child’s health in Pakistan and assess the interaction effect of maternal education and employment status.

## 2. Methods

### 2.1. Data Source

The lists of households for the country’s sampling zones were provided by the Pakistan Bureau of Statistics. It was determined that the sample size of 16,240 households—of which 7980 were in urban areas, and 8260 were in rural regions—would offer reasonable accuracy for the survey indicators. The sampling process required two stages. Using a systematic selection technique, primary sample units of 580 were taken in the first stage of sampling. Stage two of the sampling process involved selecting 28 households at random from each cluster using a systematic sampling technique based on the same probability for each. In 2017–2018, a total of 50,495 married women between the ages of 15 and 49 were subjected to an interview. In a sample of 50,495 mothers, around 2264 filled out questionnaires about their child’s health. When necessary, we deleted records one by one using a list-based deletion strategy. If any value was missing in this method, the entire record was skipped. After excluding the participants whose data was lacking, the investigation focused on analyzing the information pertaining to 2244 females.

### 2.2. Measurement

CH = Child health measured through child’s weight at birth. This variable was used as the dependent variable in our research. According to the World Health Organization, the average weight of a baby born at 37 to 40 weeks should range from 2.5 to 4.0 kg [25]. We have used the WHO criteria to measure the child’s health. It has been described in two categories. If a child’s weight at birth is less than 2.5 kilograms, then it is coded as 1; if the child’s weight at birth is at least 2.5 kilograms, then it is coded as 2.

MA = Mother’s age is divided into different groups, i.e., 15–19, 20–24, 25–29 years, and so on. 

MED = Mother’s education is divided into four categories. If the mother has no education, then it is coded as 0. If the mother has a primary education, then it is coded as 1, and so on. This variable has been categorized into two categories for the analysis of moderation. It is coded as 1 if the mother has no education and primary education and coded as 2 if the mother has at least secondary education.

MES = Mother’s employment status was measured through “Have women currently employed in the last 12 months?” Mother’s employment status is divided into two categories: it is coded as 0 if mothers are currently not employed, but if mothers are currently employed, then it is coded as 1.

WSH = Household wealth status was created by utilizing the information of the household’s residence and their asset attributes. For all assets, a score was given to every household, and a total was obtained for each of them. Every individual is placed as per the scores of the households according to their residence. It is distributed into five different quintiles, from poorest to richest. Mothers from the poorest, poorer, and middle quintiles are considered the poorest and coded as 1, and women belonging to the richer and richest quintiles are considered the richest and coded as 2.

EMM = The presence of television (TV) in households has been used as a proxy for exposure to mass media. If the household owns a TV then it is coded as 1, otherwise 0.

FED = Father’s education level is divided into two categories. If the father has no education and has attended primary school (less than secondary), then it is coded as 1. If the father has completed secondary school education and has completed higher education (at least secondary), then it is coded as 2.

BS = Birth spacing has been divided into two categories. If two consecutive births were spaced for less than 33 months, then it is coded as 1, and if spaced for at least 33 months, then it is coded as 2. 

UMNFP = The UMNFP is described as “the percentage of married women who do not use any contraceptives but wish to delay their next gestation or who do not wish to have any more children”. It has been categorized into two groups. If women have a UMNFP (for spacing and limiting) then it is coded as 1, otherwise it is coded as 0.

To examine the association between UMNFP and child health, the model used was:

CH = f(MA, MED, MES, WSH, EMM, FED, BS, UMNFP, WSH*MED, WSH*MES)

The descriptive statistics presented the frequencies and percentage of different factors of the respondents. The outcome variable was categorical, i.e., the child is not healthy, and the child is healthy. In this situation, binary logistic regression was used to investigate the causal factors related to child health. We studied the moderating effect via bootstrap-based Hayes’ PROCESS macro [26]. It is a well-known statistical technique of resampling that approximation the factors of the model and their standard errors strictly from the sample [27]. All analyses were carried out using SPSS version 24. 

## 3. Results

The socio-demographic and economic characteristics of respondents are presented in Table 1. Data from 2244 respondents were investigated. Of these, the mainstream of the mothers belongs to the age group of 25–29 years old, and more than three quarters (81.1%) of the mothers were 35 years of age. The majority women (81.6%) had their child born healthy. More than two thirds of the women had completed at least secondary school education (71.4%); three quarters of all fathers (79.4%) had received at least secondary education. Almost two thirds of the mothers (65%) belonged to the richest households, and 83.6% of the mothers were not currently employed. The data show that the majority of the mothers had EMM (82.3%), and had less birth intervals than 33 months (72.7%). The prevalence of UMNFP has been noted among 516 (23%) women, whereas 1728 (77%) women did not have any UMNFP.

The results of regression are reported in Table 2. The result of binary logistics regression showed that the child health of young mothers aged between 25 to 39 years is better as compared with that of older mothers. The odds ratio indicates that the child health of more highly educated parents is better than that of uneducated parents. The health of those children who belong to the wealthier households is better than those who belong to the poorer households. The child health of those mothers who have exposure to mass media is better as compared with that of those mothers who have no exposure to mass media. The results indicate that the child health of those mothers who are able to maintain adequate birth spacing (33 months) is better than that of those mothers who do not have proper birth spacing. The child health of unemployed mothers is better as compared with that of women who are currently employed. In other words, the child health of employed mothers is poorer than that of currently unemployed mothers. Similarly, the child health of those mothers who have UMNFP is poorer than that of those mothers who have no UMNFP. We conclude that the effect of the mother’s age (25 to 39 years) her education (higher level), father’s education (higher level), family’s wealth status (the richest), exposure to mass media, adequately spaced births (at least a gap of 33 months) on child health is positive. The effect of the MES (currently employed) and UMNFP on a child’s health is negative.

The outcomes of moderation have been reported in Table 3. There is insignificant and positive effect between education of mother and child’s health. But, we found significant and positive moderation between the household wealth status and the mother’s education on child’s health (β = 0.3498, *p* < 0.05). 

We draw the moderation effect to indicate the interacting effects of low and high mother’s education. As shown in Figure 1, wealth status and child’s health curves are much more clear and positive when the mother’s education is higher. On the other hand, we can say that a child’s health is better when the mother is wealthier and highly educated. 

The moderating effect of a mother’s employment on the relationship between the wealth status of the household and child health is also positive (β = 0.4916, *p* < 0.05). To clearly show the moderating effects of an MES (currently employed and currently unemployed), we plotted the moderation effects. As shown in Figure 2, the link between wealth status and a child’s health is clearly well defined and positive when the mother is employed rather than unemployed. We can say that the child health of those mothers who belong to wealthier families and are currently employed is better.

## 4. Discussion

Mother’s education, MES, wealth status of a household, and UMNFP have been identified as major determinants of child survival in Pakistan. A positive correlation exists between a mother’s age and her child’s health. Older mothers use maternal health care services more frequently and their infants are healthier than those of younger mothers [28,29,30,31,32,33]. It is to be noted that parental education and a child’s health are positively associated. Previous studies proved that a well-educated mother could take better care of her health during pregnancy and take care of her child in a better way than a less educated mother [34,35,36,37,38]. The second reason for this positive relation is that educated mothers are more likely to use informed health knowledge and practices that are important for their children’s health [39,40,41,42,43]. Similarly, a father’s education can also enhance the probability of a child’s survival because the father’s education plays a vital role in providing monetary support to family and basic health facilities in health crises [44,45,46]. 

We found an inverse association between women’s employment and child health. The employment rate among mothers with children under the age of 18 months has increased considerably (reaching 71.2% in 2018) [47]. Earlier studies recommend that a mother’s employment is associated with child health care [48,49,50,51,52,53,54]. Children under the age of 15 months whose mothers are employed full-time have 0.18 fewer precautionary child health visits (immunization visits) annually compared with those of mothers who are not working [51,52,55,56]. A household’s wealth status indicates the population’s income level and living standards. Wealthier households are more likely to have children with better health than those from poorer households [57,58,59,60,61,62]. Household poverty has been stated to reduce child health by reducing access to adequate healthcare [63]. It is found that there is a high probability of a child’s health being better if their mother has access to television. This is because television programs provide information regarding proper health facilities and child vaccination that are compulsory for children [20,64]. Birth spacing is related to both children’s health and childhood mortality [65,66], and even in developed economies, being poor during pregnancy has long-term effects on socioeconomic success [67,68] and health [69,70,71,72,73]. A child’s survival is affected by the spacing between births. According to a prior study, when the interval between births is less than two years, the risk of infant mortality is, on average, doubled [74,75]. There is a direct association between UMNFP and child health. Due to UMNFP, the birth spacing between two children is lower, and the health of newborn babies will be poor. 

The moderating effect of maternal education on the relationship between household wealth status and child health is also positive. This indicator is taken because a child will be healthy during pregnancy or in early life if they are born to a well-educated and financially stronger family. As well as this, the mother will probably have all the necessary health facilities and be well aware of child care seeking during a pregnancy. The result of moderation shows that a mother’s employment status strongly links household wealth status and child health. Both slopes are significantly different from zero, and the mother’s employment slope is negative, but the household wealth status slope is positive and illustrates that in the presence of employment, the relationship between household wealth status and child health is more protective and healthier. On the other hand, we can say that child health is better when the mother’s socioeconomic status (education, wealth status, and employment status) is better. 

## 5. Recommendations

According to the findings of this article, it will be essential to steadily and significantly boost health and education spending if we are to reduce infant and maternal deaths and meet the Sustainable Development Goals by 2030. Employment and education increase knowledge and self-assurance, enhancing the standard of living and health of the family and society as a whole. The role of the digital space and mass media is significant. Therefore, the government of Pakistan should provide means to support girls’ education and launch various programs to support households’ efforts to control their families’ size for them to be able to provide a better standard of life for their children and fulfill the role of engaged parents to the very best of their abilities. To protect both the mother’s and the child’s health, governments should place more emphasis on expanding access to birth control methods and raising public knowledge of the importance of adequate birth intervals. To raise understanding about the significance of birth spacing, it is absolutely necessary for the government to make contraceptives available through community outreach programs and the public media. There should be regular research on family planning and mother and child health so that problems can be found sooner and the right programs can be implemented.

## 6. Limitations and Direction for Further Study

The objective of the study was to investigate the moderating role of maternal education and employment status on the relationship between wealth status and child health with respect to Pakistan. The limitation of the current study is that we did not test the impact of other predictors and moderating variables in the proposed relationship. Notably, child health can also be affected by a number of other determinants which need to be examined. For example, a father’s education and employment status can also play an interactional role in the relationship between the wealth status of the household and the child’s health because a father’s education plays a crucial role in creating awareness about healthy and nutritious food, which is beneficial for child health. He will be vigilant in keeping children away from contaminated and preservative-laden food. Moreover, the father’s employment status is also essential in offering monetary support to the family and basic health facilities at the time of health crises.

## 7. Conclusions

To conclude, a mother’s education, MES, wealth status of the household, and UMNFP have been found as major determinants of child health. The results of this study show that there is a decisive necessity to grab social factors affecting maternal and child health in Pakistan, which narrate important problems of the mother’s status, employment, education, and participation in decision-making. The role of a mother’s education in improving child survival is a dire necessity for girls’ education investments. Integrating health and expanding health care messages involving child and maternal health to other areas, such as employment, education, empowerment, birth spacing, and deterrence of child marriage, are vital for the ministry of health in tandem with the provinces. Governments should also pay attention to providing education to women, both formal and informal, so that they can act in a better way in their personal lives as well as in the development of the nation. The involvement of mass media to increase awareness regarding the advantages of family planning and birth spacing, especially for women’s and children’s health, is crucial for the country, through both the public and private sectors.

## Figures and Tables

**Figure 1 children-09-01559-f001:**
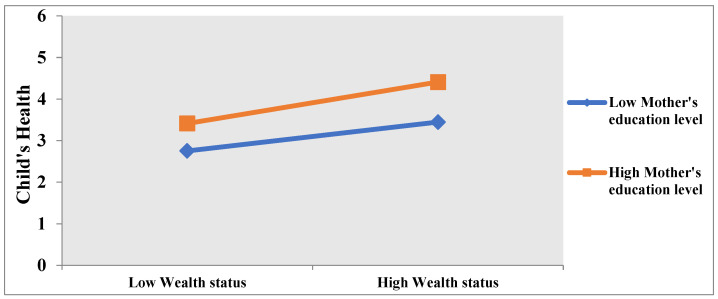
Moderation effect of mother’s education and wealth status of household on child health.

**Figure 2 children-09-01559-f002:**
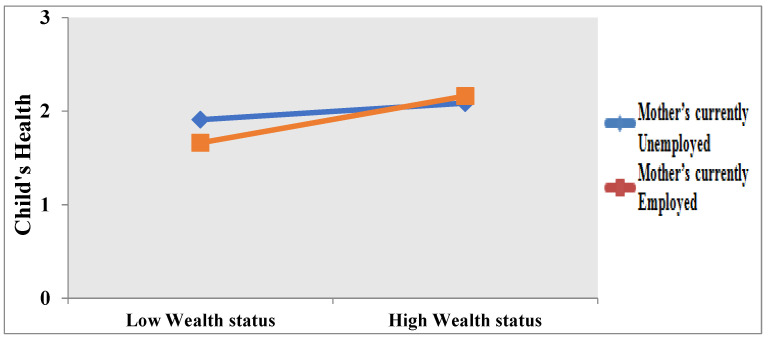
Moderation effect of mother’s employment status and wealth status of household on child health.

**Table 1 children-09-01559-t001:** Socio-economic and demographic determinants of respondents.

Socio-Economic Characteristics		Frequency	Percentage (%)
Child’s health	Unhealthy	414	18.4
	Healthy	1830	81.6
Mother’s age	15–19	44	2.0
	20–24	400	17.8
	25–29	731	32.6
	30–34	645	28.7
	35–39	323	14.4
	40–44	89	4.0
	45–49	12	0.5
Mother’s education	No education	387	17.2
	Primary	254	11.3
	Secondary	720	32.1
	Higher	883	39.3
Mother’s employment status	Unemployed	1936	86.3
	Employed	308	13.7
Wealth status of household	Poorest	785	35.0
	Richest	1459	65.0
Exposure to mass media	No	397	17.7
	Yes	1847	82.3
Father’s education	No education	239	10.7
	Primary	223	9.9
	Secondary	856	38.1
	Higher	926	41.3
Birth spacing	Less than 33 months	1631	72.7
	At least 33 months	613	27.3
Unmet need for family planning	No	1728	77.0
	Yes	516	23.0

**Table 2 children-09-01559-t002:** Results of binary logistic regression.

Independent Variables		Β	Sig.	Odd Ratios	Class Intervals
Constant		0.152	0.223	1.165	-------
Mother’s age	15–19	Reference			
	20–24	0.585	0.089	1.795	1.259–1.934
	25–29	0.653 *	0.049	1.921	1.721–2.136
	30–34	0.734 *	0.031	2.084	1.973–2.249
	35–39	0.827 *	0.021	2.287	1.857–2.513
	40–44	0.697	0.104	2.009	0.674–2.105
	45–49	0.809	0.343	2.245	0.368–2.843
Mother’s education	No education	Reference			
	Primary	0.109	0.589	1.115	0.721–1.486
	Secondary	−0.042	0.801	0.958	0.683–1.135
	Higher	0.502 **	0.008	1.652	1.298–1.991
Mother’s employment status	Employed	Reference			
	Unemployed	0.315 *	0.046	1.370	1.122–1.537
Wealth status of mother’s household	Poorest	Reference			
	Richest	0.217 *	0.041	1.242	1.092–1.411
Exposure to mass media	No	Reference			
	Yes	0.030 *	0.032	1.160	1.059–1.253
Father’s education	No education	Reference			
	Primary	−0.186	0.409	0.831	0.627–1.092
	Secondary	0.020	0.916	1.020	0.759–1.108
	Higher	0.087 *	0.042	1.091	1.013–1.213
Birth spacing	Less than 33 months	Reference			
	At least 33 months	0.120 *	0.024	1.128	1.051–1.257
Unmet need for family planning	Yes	Reference			
	No	0.112 *	0.027	1.112	1.028–1.261

** *p* < 0.01; * *p* < 0.05.

**Table 3 children-09-01559-t003:** Results of moderation.

Independent Variables	β	Sig.
Mother’s education	0.3241	0.401
Mother’s employment status	−0.9363 *	0.049
Wealth status of household	0.2861 *	0.045
Wealth status of household * Mother’s education	0.3498 *	0.037
Wealth status of household * Mother’s employment status	0.4916 *	0.012

* *p* < 0.05.

## Data Availability

We have used the secondary data of PDHS 2017–2018. Available at: https://www.nips.org.pk/study_detail.php?detail=MTgw (accessed on 13 May 2021).

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
