# Peer review of "The Moderating Role of Maternal Education and Employment on Child Health in Pakistan"

_children, 2022, doi:10.3390/children9101559_

Round 1
Reviewer 1 Report
The authors used publicly available 2017-2018 Pakistan Demographic and Health Survey (PDHS) data to examine the effect of unmet need of family planning (UMNFP) and the moderating effect of maternal education and employment status on child health. However, there are serious problems in this manuscript. First, the term "child health" used is in this study only means birthweight status, not what readers imagine as "child health" . They should use "birthweight" instead, in the title and the text. Second, the description of PDHS is insufficient. Some readers may not be familiar with this survey, so please provide a brief outline of the survey in the Methods, such as subject sampling, data collection methodology, and survey area. Third, I could not understand why important variables such as maternal parity (or the number of children in the family), gestational length (or whether the child was born preterm or not), singleton or multiple pregnancies, urban rural differences, and number of prenatal visits. I also could not figure out whether the above information was available in the survey data or not. There are no explanations on primary education in Pakistan, so I could not understand whether the current categorization is appropriate or not. Explanation on employment status is vague. What if the family owned a business? Wealth status was categorized based on quintles of what? Please provide more information. I think the figures shown in Table 1 is opposite for poorest and richest. I think the most serious thing is that the definition of UMNFP is not clear. Please provide more information on how this was assessed.
Finally, explanation is missing on why the authors selected only maternal factors for the moderation analysis. Did you check the interactions with other paternal factors, such as father' employment status? I think the authors should check their dataset more thoroughly.
Author Response
The authors used publicly available 2017-2018 Pakistan Demographic and Health Survey (PDHS) data to examine the effect of unmet need of family planning (UMNFP) and the moderating effect of maternal education and employment status on child health. However, there are serious problems in this manuscript. First, the term "child health" used is in this study only means birthweight status, not what readers imagine as "child health". They should use "birthweight" instead, in the title and the text.
Response: Thank you for your valuable comment. Many of previous published articles birthweight has been used as a proxy of child health. For example the studies of Asif et al. (2022), Spencer (2017), and Manyeh et al. (2016).
Second, the description of PDHS is insufficient. Some readers may not be familiar with this survey, so please provide a brief outline of the survey in the Methods, such as subject sampling, data collection methodology, and survey area.
Response: Thank you for valuable comment. Please see data source section on page 6.
Third, I could not understand why important variables such as maternal parity (or the number of children in the family), gestational length (or whether the child was born preterm or not), singleton or multiple pregnancies, urban rural differences, and number of prenatal visits. I also could not figure out whether the above information was available in the survey data or not.
Response: The information of above variable is included in PDHS. But first objective of our study is to investigate the effect of UMNFP on child health, and second objective of our study is to investigated the moderating effect of mother’s education and their employment on child health.
There are no explanations on primary education in Pakistan, so I could not understand whether the current categorization is appropriate or not.
Response: This categorization of the variable has been categorized by Pakistan Demographic and Health Survey. This survey was conducted by Pakistan Bureau of Statistics with the collaboration of USAID.
Explanation on employment status is vague. What if the family owned a business?
Response: Mother’s employment status has been measured through “Have women currently employed in the last 12 months?”
Wealth status was categorized based on quintiles of what? Please provide more information. I think the figures shown in Table 1 is opposite for poorest and richest.
Response: Thank you for valuable suggestion. Comment incorporated please see the last three lines of page 7, and the figure of wealth status shown in Table 1 is corrected, we again verify from data after removing the missing data.
I think the most serious thing is that the definition of UMNFP is not clear. Please provide more information on how this was assessed.
Response: Thank you for your valuable comment. Please see measurement section on page 8.
Finally, explanation is missing on why the authors selected only maternal factors for the moderation analysis. Did you check the interactions with other paternal factors, such as father' employment status? I think the authors should check their dataset more thoroughly.

Reviewer 2 Report
The moderating role of maternal education and her employment on child's health in Pakistan (children-1867712)
(Review)
Main message of the article
In the manuscript entitled “The moderating role of maternal education and her employment on child's health in Pakistan”, the authors analyzed a publicly available dataset to investigate the effect of unmet need for family planning on a child’s health together with the moderating role of mother’s education and employment status on the child’s health. Results provide insight on the factors having an impact on child’s health in Pakistan.
General Judgment Comments
The article is clearly written and interesting. While the title provides enough information to frame the article, I would recommend the authors to add new keywords for helping the manuscript to gain visibility. The abstract is clear but needs more work on the definition of concepts and on the reporting of results. The statistical analysis is appropriate, and the sample size has enough statistical power. Figures and Tables are missing the caption. Furthermore, from the Figures it is not clear whether any statistically significant result emerged. Figures also need more work on the graphical side, such as optimizing the y axis (in terms of label and range of interest) and reducing the white space around the lines.
I would recommend the manuscript to undergo Major Revision.
Major Issues
-
- In the abstract, it is not clear what the unmet need for family planning is. Please define the concept.
-
- In the abstract, results are not reported following the standard format. Are there any significant results in the analysis?
-
- The Abstract section would benefit from a sentence of conclusion in which the authors provide the significance of the results.
-
- Some Keywords needs to be added to correctly frame the study in the literature.
-
- Some other works are relevant to frame motherhood and mother’s mental health in Pakistan:
â–ª Neoh, M. J. Y., Airoldi, L., Arshad, Z., Bin Eid, W., Esposito, G., & Dimitriou, D. (2022). Mental Health of Mothers of Children with Neurodevelopmental and Genetic Disorders in Pakistan. Behavioral Sciences, 12(6), 161;
â–ª Maselko, J., Sikander, S., Bangash, O., Bhalotra, S., Franz, L., Ganga, N., ... & Rahman, A. (2016). Child mental health and maternal depression history in Pakistan. Social psychiatry and psychiatric epidemiology, 51(1), 49-62.
-
- Figure 1 and Figure 2: figures do not include a caption in which the authors explain the significance of the represented results. From both Figures, it is not clear whether the represented results are significant or not. Furthermore, in both Figures, the labels on the y axis are almost not readable and I would suggest removing some values and keeping only the 3-4 relevant values. The white space in the graph should also be reduced. Finally, for the reader it might not be clear whose wealth status the authors are representing.
-
- How did the authors remove missing data?
-
- Please provide the demographic details of the sample in the Methods (subsection Participants).
-
- The Measurement section should not be written in a list-format. Please explain the equation with a non-list text.
-
- To help the reader, I would suggest the authors to first introduce all the variables of the study and only after to present and explain the
functional form of the model.
-
- “If a child's weight at birth is less than 2.5 kilograms then coded as 1 and if the child's weight at birth at least 2.5 kilograms then coded as
2”. Why did the authors use 2.5 Kg as value of reference for the weight?
-
- Please provide information on how the dataset was originally collected.
-
- Tables should have a caption.
-
- Throughout the manuscript, results are often not reported following the standard format. Please adjust.
Minor Issues
-
- What kind of universal responsibilities do the authors refer to in the first line of the abstract?
-
- Abstract: “We analyzed the data of 2,244 women in this study” this sentence should go in the Methods of the abstract.
Final comments
I would recommend the manuscript to undergo Major Revision.
Author Response
General Judgment Comments
The article is clearly written and interesting. While the title provides enough information to frame the article, I would recommend the authors to add new keywords for helping the manuscript to gain visibility. The abstract is clear but needs more work on the definition of concepts and on the reporting of results. The statistical analysis is appropriate, and the sample size has enough statistical power. Figures and Tables are missing the caption. Furthermore, from the Figures it is not clear whether any statistically significant result emerged. Figures also need more work on the graphical side, such as optimizing the y axis (in terms of label and range of interest) and reducing the white space around the lines.
Response: Thank you for appreciation. All comments have been incorporated.
Major Issues
In the abstract, it is not clear what the unmet need for family planning is. Please define the concept.
Response: Thank you for suggestion. Comment incorporated. Please see the 8 and 9 lines on page 2.
In the abstract, results are not reported following the standard format. Are there any significant results in the analysis?
Response: Comment incorporated. Please see the results section of abstract.
The Abstract section would benefit from a sentence of conclusion in which the authors provide the significance of the results.
Response: Comment incorporated. Please see the conclusion section of abstract.
Some Keywords needs to be added to correctly frame the study in the literature.
Response: Thank you for suggestion. Comment incorporated.
Figure 1 and Figure 2: figures do not include a caption in which the authors explain the significance of the represented results. From both Figures, it is not clear whether the represented results are significant or not. Furthermore, in both Figures, the labels on the y axis are almost not readable and I would suggest removing some values and keeping only the 3-4 relevant values. The white space in the graph should also be reduced. Finally, for the reader it might not be clear whose wealth status the authors are representing.
Response: Both figures are updated and reduced the white space.
How did the authors remove missing data?
Response: Thank you for suggestion. Comment incorporated. Please see data source section.
Please provide the demographic details of the sample in the Methods (subsection Participants).
Response: Thank you for suggestion. Comment incorporated. Please see data source section.
The Measurement section should not be written in a list-format. Please explain the equation with a non-list text.
Response: Comment Incorporated.
To help the reader, I would suggest the authors to first introduce all the variables of the study and only after to present and explain the functional form of the model.
Response: Functional form of the model is presented after the explanation of the variables.
“If a child's weight at birth is less than 2.5 kilograms then coded as 1 and if the child's weight at birth at least 2.5 kilograms then coded as 2”. Why did the authors use 2.5 Kg as value of reference for the weight?
Response: Child weight at birth has been used as a proxy of child health. According to World Health Organization, average weight of a baby born at 37 to 40 weeks should ranges from 2.5 to 4.0 kg (WHO, 2007). We have used the WHO criteria to measure the child’s health.
Please provide information on how the dataset was originally collected.
Response: Comment incorporated. Please see the section of data source.
Methods must be adequately described.
Response: Comment incorporated.
Tables should have a caption.
Response: Comment has been incorporated.
Throughout the manuscript, results are often not reported following the standard format. Please adjust.
Response: Results are reported according to journal requirements.
What kind of universal responsibilities do the authors refer to in the first line of the abstract?
Response: Comment incorporated. Please see the first and second lines of abstract.
Abstract: “We analyzed the data of 2,244 women in this study” this sentence should go in the Methods of the abstract.
Response: Comment incorporated. Please see line 12 page 2.

Round 2
Reviewer 2 Report
The authors have responded to the questions
Author Response
Finally, explanation is missing on why the authors selected only maternal factors for the moderation analysis. Did you check the interactions with other paternal factors, such as father' employment status? I think the authors should check their dataset more thoroughly. May the authors add a comment on this among the limitations of the study?
Response: Thank you for suggestion. We did not test the moderating role of father employment and education in the relationship between the wealth status of the household and the child's health because it was not the scope of the study. However, we mention this as a limitation and direction for further study in the revised manuscript.